# Tunable Fluorescence via Self-Assembled Switching of AIE-Active Micelle-like Nanoaggregates

**DOI:** 10.3390/ijms24129941

**Published:** 2023-06-09

**Authors:** Amal Farghal Noreldein Elsyed, Gah-Lai Wong, Mohamed Ameen, Min-Wei Wu, Cheng-Chung Chang

**Affiliations:** 1Graduate Institute of Biomedical Engineering, National Chung Hsing University, 145 Xingda Road, Taichung 402, Taiwan; amalnour2390@gmail.com (A.F.N.E.); kallywong@cs.nchu.edu.tw (G.-L.W.); mohamed.ameen.chem@gmail.com (M.A.); miss212518@gmail.com (M.-W.W.); 2Intelligent Minimally-Invasive Device Center, National Chung Hsing University, Taichung 402, Taiwan

**Keywords:** Suzuki reaction, self-assembly, locally excited (LE) state, intramolecular charge transfer (ICT), aggregation-induced emission enhancement (AIEE), structure–activity relationship (SAR)

## Abstract

Chemical structures bearing a combination of aggregation-induced emission enhancement (AIEE) and intramolecular charge transfer (ICT) properties attracted the attention of many researchers. Recently, there is an increasing demand to pose tunable AIEE and ICT fluorophores that could present their conformation changes-related emission colors by adjusting the medium polarity. In this study, we designed and synthesized a series of 4-alkoxyphenyl-substituted 1,8-naphthalic anhydride derivatives NAxC using the Suzuki coupling reaction to construct donor–acceptor (D-A)-type fluorophores with alkoxyl substituents of varying carbon chain lengths (x = 1, 2, 4, 6, 12 in NAxC). To explain the observation that molecules with longer carbon chains revealed unusual fluorescence enhancement in water, we study the optical properties and evaluate their locally excited (LE) and ICT states by solvent effects combined with Lippert–Mataga plots. Then, we explored the self-assembly abilities of these molecules in water-organic (W/O) mixed solutions and observed the morphology of its nanostructure using a fluorescence microscope and SEM. The results show that NAxC, x = 4, 6, 12 show different degrees of self-assembly behaviors and corresponding aggregation-induced emission enhancement (AIEE) progresses. At the same time, different nanostructures and corresponding spectral changes can be obtained by adjusting the water ratio in the mixed solution. That is, NAxC compounds present different transitions between LE, ICT and AIEE based on the polarity, water ratio and time changes. We designed NAxC as the structure–activity relationship (SAR) of the surfactant to demonstrate that AIEE comes from the formation of micelle-like nanoaggregates, which causes a restriction of the transfer from the LE state to the ICT state, and micelle formation results in a blue-shift in emission and enhances the intensity in the aggregate state. Among them, NA12C is most likely to form micelles and the most obvious fluorescence enhancement, which will switch over time due to the nano-aggregation transition.

## 1. Introduction

Local excitation (LE), intramolecular charge transfer (ICT) and aggregation-induced emission enhancement (AIEE) are three distinct excited state phenomena that have been previously reported in many articles [1,2,3,4]. The nanomaterials that are characterized by any of these properties are important and have been used in a wide range of applications such as biomedical applications, fluorescent probes, biosensors, biomarkers, OLEDs, chemo-sensors, photovoltaic devices, etc. [5,6]. Since the discovery of the donor-acceptor (D-A) DMABN by Lippert, which is dual-fluorescence in a highly polar solvent [7], the charge transfer phenomenon was highlighted by researchers. ICT typically occurs when molecular structures construct with both electron-pushing (donor) and electron-pull (acceptor) groups; such molecules are often sensitive to the environment through variations in their photophysical properties. Therefore, we usually use the spectral changes derived from the solvent effect to examine the ICT characters of molecules and evaluate their potential as solvatochromic probes [8]. However, the strong ICT effect usually quenches the emission of the fluorophore [9], and, mostly, the D-A fluorophores are facing the problem of aggregation-caused quenching (ACQ), which also limited their applications in solid thin films, such as biophotonic and electroluminescence areas [10]. AIEE has recently overcome the problem of ACQ and opened the door for many fluorophores to be applied as solid materials for different applications. It was explained that the aggregation of these molecules will prevent the twisting intramolecular charge transfer (TICT) or restrict the intramolecular rotation in the free parts [11]. Although it is difficult to have materials with a combination of AIE and ICT, in recent years, some materials with this combination have been reported [7,9,11,12].

Materials with intramolecular charge transfer (ICT) and aggregation-induced emission enhancement (AIEE) properties have garnered significant attention in recent years due to their exciting prospects for developing advanced functional materials with enhanced optical properties, improved performance and unique optical properties, which give it potential for applications in areas such as optoelectronics, sensing, bioimaging and light-emitting devices. For example, Ruixue Z. et al. demonstrated the design and synthesis of DBABN, a donor-acceptor compound with efficient ICT and AIEE, which has been used as a turn-on fluorescent sensor for methanol vapor [13]. Moreover, researchers have made significant progress in tailoring the molecular structure and packing motifs to achieve desirable ICT and AIEE properties. The incorporation of conjugation, the introduction of electron-donating and electron-accepting groups and the control over intermolecular interactions have emerged as effective strategies for modulating the charge transfer and emission behavior of these materials. For instance, Yahui C. et al. reported the design of a novel series of compounds with a strong AIEE and ICT character, and the combination of those compounds gives a white light emission in various solvents, enabling its utilization as a fluorescent probe for organic solid emitters and white-light-emitting materials [14]. Kamran H. et al. produced AIE materials with a strong ICT effect that are able to generate Singlet oxygen (^1^O_2_); thus, they were good candidates for photodynamic therapy and lipid droplets detecting in living cells [15].

Micellar aggregates are often formed by the self-assembly of molecules constructing both hydrophilic and hydrophobic moieties within one molecule [16]. Recent decades have seen an increased interest in imaging biological tissues using fluorophore micelles. In the early days, fluorescent micelles were often fabricated by encapsulating fluorophore within the micelle cores, which is a kind of co-assembly micelle [17,18,19]. In these cases, not only must the CMC of co-assemblies be re-evaluated, but it may also interfere with the inclusion of another molecule [17,20]. Recently, fluorescent micelles that are emissive unimolecular assemblies have offered a clear advantage in resolving the problem above. It may permit the enclosure of additional hydrophobic molecules in the remaining core or construct a fluorescent micelle with a fluorescent surfactant [21,22,23]. Thus, much effort has been devoted to the development of luminescent micelles to achieve both carrier ability and emissive features. In previous studies, we applied AIEE properties to describe the fluorescence behavior of the solid state and water-soluble fluorogen and developed a fluorescent organic nanoparticle (FON) model based on a surfactant-like fluorophore, which can further self-assemble to form micelle-like nanoaggregates due to its amphiphilic characteristic. In the aggregation state, intramolecular rotation is restricted, emission is thus greatly enhanced and, eventually, the AIEE phenomenon is the cause of FONs [24,25,26,27].

In this manuscript, we present the absorption and photoluminescence features of a series of compounds NAxC, composed of 1,8-naphthalic anhydride chromophores substituted with 4-alkoxylphenyl moiety groups. Solvatochromic studies of the A–D type chromophore NAxC were performed to calculate the Lippert–Mataga plot and determine the dipole moments exchange to evaluate the intrinsic ICT properties. Then, the AIEE properties were investigated, and these phenomena were applied to illustrate the unusual luminescence of molecules in an aqueous solution. The morphology and the stacking formation transfer of the nanostructures in the water–organic solvent mixed system were also observed by SEM and fluorescence microscope. On the other hand, we discussed that some molecules of NAxC presented the ability to form micelles, and then the micelle-like nanoaggregate was the main cause of AIEE. Eventually, the relationships between the tunable emissions and the nano-aggregates shapes in the water–organic system can be explained by the transitions between LE, ICT and AIEE based on the polarity, water ratio and time changes.

## 2. Results

### 2.1. Design and Optical Properties of 1,8-Naphthalic Anhydride Derivatives

1,8-naphthalic anhydride has shown good optoelectronic properties and has been applied in different applications [28,29]. More important, it has been used as an intermediate core to synthesize different derivatives of 1,8-naphthalimide [24,30]. In this investigation, the 1,8-naphthalic anhydride moiety was selected as the π-electron acceptor because of its good chemical stability and the commercial availability of various useful molecular building blocks. The 4-alkoxyphenyl group was chosen as the electron-donating unit for constructing an A–D type chromophore with 1,8-naphthalic anhydride by the Suzuki coupling reaction. Figure 1 showed that 4-alkoxyphenyl substituted 1,8-naphthalic anhydride derivatives (products NA1C, NA2C, NA4C, NA6C, NA12C), as prepared in this manuscript.

The solvent polarity effect was investigated by collecting the absorption and fluorescence of compounds in different solvents (H_2_O, MeOH, EtOH, Acetone, DMF, DMSO, CH_2_Cl_2_, THF, EA, and Toluene). As shown in Appendix A, all compounds showed similar absorption wavelengths (λ_abs_~370 ± 5 nm), with slight changes in the absorption values observed in all the solvents except for H_2_O, whereby the absorption values decreased and the peak wavelengths shifted to 385 ± 4 nm. This indicates that, in their ground states, these chromophores are not significantly stabilized by solvation due to a relatively small dipole moment. On the other hand, as shown in Figure 1a, strong solvatochromic redshifts with decreasing emission intensities of the emission spectra were observed for all five compounds with increasing solvent polarity. For example, in less polar solvents (e.g., Toluene, EA, THF and CH_2_Cl_2_), the compounds showed high fluorescence intensities, with the highest fluorescence intensity observed in Toluene. The wavelengths of the maximum absorption and emission, molar absorptivity, quantum yields and Stokes shifts of all NAXC compounds in different organic solvents, comparing a wide polarity range, are listed in Table 1. Interestingly, when we discuss the fluorescence emission spectra of these NAXC derivatives in water, a notable decrease in the fluorescence intensity of NA1C was observed.

Upon increasing the carbon chain length, NA2C, NA4C, NA6C and NA12C show a carbon chain-dependent increase in the emission intensities, with an obvious blueshift, with respect to other polar solvents. Figure 1b shows the solvent effect-related fluorescent emission change in the color of 10 μM compounds. This series of NAXC compounds differ only in the carbons number and were designed to offer the same push–pull effect. We first discuss the solvatochromism relative intramolecular charge transfer (ICT) via Lippert–Mataga plots (1) [24,31], which describes the dependence of the energy difference between the ground state and the excited state (in cm^−1^) on the refractive index (*n*) and the dielectric constant (*ε*) of the solvent: (1)$$\Delta \nu =\nu_{abs}-\nu_{em}=\frac{2(\mu_{e}-\mu_{g}{)}^{2}}{hca^{3}}\Delta f+\mathrm{constant}$$
where Δ*ν* stands for the Stokes shift, *h* is Planck’s constant, *c* is the speed of light, *a* is the radius of the cavity in which the fluorophore resides, *ν_abs_* and *ν_em_* are the wavenumbers (cm^−1^) of the absorption and emission and *μ_g_* and *μ_e_* are dipole moments in the ground state and the excited state, respectively. Δ*f* is the orientation polarizability and can be defined as Equation (2).
(2)$$\Delta f=f\left(\varepsilon \right)-f\left(n^{2}\right)=\frac{\left(\varepsilon -1\right)}{\left(2\varepsilon +1\right)}-\frac{\left(n^{2}-1\right)}{\left(2n^{2}+1\right)}$$
where *ε* is the dielectric constant of the medium and *n* is the refractive index of the solvent. The slope (Δ*ν* vs. Δ*f*) of the Lippert plot reflects the solvent sensitivity of a molecule and is presented in Appendix A; calculated results for each molecule are presented in Table 1. Appendix A shows that a positive solvatochromism was found for all chromophores, indicating the involvement of solvent polarity-dependent intramolecular charge transfer (ICT) emissive states [32,33]. That is, molecules in the excited state are polarized by a higher dipole moment compared to that in their ground state, as shown in Table 1.

### 2.2. Self-Assembly of NAXC

Following the Lippert–Mataga plots result, we expected that NAXC in H_2_O should be the most red-shift, with very low emissions. However, as shown in Table 1 and Figure 1, some of these molecules not only deviate from Lippert plots but even exhibit fluorescence emissions in the aqueous solution. It is known that there are not many organic molecules that exhibit significant fluorescence in an aqueous medium. To study the special spectral behaviors of NAXC, we must first notice that the absorption energy of the compound decreases while the fluorescence emission intensity increases, as the carbon number of the substituted alkoxy chain increases in NAXC (Figure 2a). Furthermore, as described in the introduction, the surfactant-like fluorophore was expected to self-assemble to form a micelle-like nanoaggregate, as the FON. We constructed the NAXC with 1,8-naphthalic anhydride as the hydrophilic head and the alkoxyl chain on the substituent as the hydrophobic tail. Accordingly, it is speculated that these compounds have the tendency to self-assemble in H_2_O to form micelles. Hence, we measured their fluorescence intensity performance at different concentrations to obtain the critical micelle concentration (CMC) of these compounds. As shown in Appendix A, the CMC values of NA2C, NA4C, NA6C and NA12C are ~1 μM, 0.5 μM, 0.2 μM and 0.05 μM, respectively. As expected, no CMC value was observed for NA1C, which is in line with its characteristic of the emission spectrum in water. Figure 2b shows the fluorescence microscope images for the compounds above CMC in H_2_O. NA2C showed a not-so-well-defined structure. NA4C, NA6C and NA12C showed round granular structures. These fluorescent structures with their shapes can verify the self-assembly property of the compounds, which in turn explains the different fluorescence behavior in H_2_O.

### 2.3. Aggregation-Induced Emission Enhancement

Most of the literature evaluates aggregation-induced emission enhancement (AIEE) by examining the fluorescence enhancement of molecules in THF/H_2_O solvent pairs. Appendix A show that upon increasing the polarity of the solutions by increasing the water fractions (*f*_W_), the absorption wavelengths did not change significantly until the water fractions were above 70%. Herein, the tail of the absorption spectrum is uplifted because of the light scattering caused by molecular aggregations [34,35]. Meanwhile, the corresponding fluorescence spectra present apparent variations. The emission peak of each molecule of NAXC starts from 480 nm and, with the increase in *f*_W_, red-shifts gradually to about 525 nm, accompanied by a decrease in the emission intensity. When the *f*_W_ reaches a certain level (>70%), the emission wavelengths of NAXC produce different degrees of blueshifts and, finally, redshifts again. We hypothesize that these different behaviors of the compounds in the mixed solvents can be attributed to the change in the solvent-pair polarity. Upon increasing H_2_O fractions, the polarity increased, which in turn gradually increases the ICT property for the D-A compounds and causes the observed redshifts [36]. On the other hand, when H_2_O fractions are 70% and above, it may lead to changes in the molecular arrangement or stacking, which include TICT switching to cause the blue shifts [37]. That is, the THF/H_2_O assay leads to the switching of molecular packing, which causes the molecular emission energy to be converted from LE to ICT and then to TICT. Nevertheless, since the fluorescence intensity decreases continuously with the increase in the water ratio, we believe that the AIEE phenomenon is not induced in the THF/H_2_O system, regardless of the molecular stacking transformation.

According to the literature [38], in the three types of excited states of the fluorophore, the TICT state has a more twisted conformation, a large twist driving force, the longest λ_max_, the broadest bandwidth, a very low quantum yield and the largest dipole moment. Compared with ICT, the LE state of the organic fluorophore should be provided with the shortest λ_max_, a narrow bandwidth, a lower quantum yield and a very small dipole moment, which is independent of the solvent polarity and temperature. To gain a deeper understanding of the excited state properties, temperature-varying experiments of the compounds NAXC were carried out, as shown in Figure 3. At a temperature of 300 K, the fluorescence spectra of the NAXC compounds in ethanol display emissions peaks at about ~520 nm. As the temperature decreases, the emission peaks undergo a blueshift to ~450 nm, which is similar to the emission wavelength in a non-polar solvent environment (Toluene). It means that the excited-state interconversion was restricted in a low temperature condition, which is the LE emission of NAXC, and 520 nm should be the ICT state.

On the other hand, the AIEE property can be investigated not only in the THF/H_2_O mixture but also in many strong/poor solvent mixtures [35]. Herein, the DMF/H_2_O mixture has been selected to verify the existence of the AIEE phenomenon. The results and discussion of the absorption spectra of NAXC in the DMF/H_2_O are similar to those in the THF/H_2_O mixture. Alternatively, the fluorescence spectra showed variable behaviors, as shown in Figure 4a. All compounds gave emission in pure DMF at ~515 nm. Upon increasing H_2_O fractions, the emission bands redshifted to ~540 nm, with a dramatic decrease in the fluorescence intensities (*f*_W_ = 50% in the case of NA1C and 40% in the case of NA2C-NA12C). After adding H_2_O, the emission bands of NA1C and NA2C were blueshifted to ~448 nm without an apparent increase in the fluorescence intensity. On the other hand, NA4C and NA6C redshifted again to ~500 nm with an increase in the fluorescence intensity upon increasing the water fractions to over 70%, respectively. Interestingly, in a system containing 50% H_2_O, NA12C already redshifts with emission enhancement and keeps constant. It seems that the system has reached the equilibrium of molecular stacking transition at 50% of the time. These energies and intensities of the fluorescent signals relative to the ratio of water and the polarities of the environments are plotted as shown in Figure 4b. For NA1C and NA2C, by increasing the H_2_O fraction, the emission signals have a two-stage change. The first stage is the redshift accompanied by a decrease in the fluorescence intensity. The second stage is the blueshift to 450 nm with no significant increase in the fluorescence intensity. NA4C and NA6C present three-stage changes. The first two stages are similar to those of NA1C and NA2C. The difference is in the third stage, where there is a redshift to 500 nm with an increase in the fluorescence intensity. The two stages of changes in the case of NA12C are as follows: a blueshift accompanied by a decrease in the fluorescence intensity was observed, followed by a redshift to 500 nm with an obvious increase in the emission intensities. We conclude that obvious redshifts accompanied by emission increases were due to the aggregates formed in a certain degree of water fractions. That is, these NA4C, NA6Cand NA12C formed self-aggregates in the DMF/H_2_O mixture, and these fluorescence data indicate the restriction of intramolecular rotations and the suppression of TICT to minimize the π–π interaction of the aromatic rings; therefore, the AIEE effect occurred instead of the ACQ.

### 2.4. Time-Dependent Color Reversible Changes of the Fluorescent Aggregates

It was found that the long-term incubation of NA12C in a DMF/H_2_O mixed solution revealed the fluorescence color changes over time. As shown in Figure 5a, in the case of 50–70%, the fluorescence changed color gradually from light-green to a blue color in 30 min with increasing emission intensities, with respect to Figure 4. The relative spectra are shown in Figure 5b, and these tunable color changes are reversible by adding and removing H_2_O, as shown in Figure 5c. SEM and fluorescence microscope images show the time-dependent nanoaggregate formation of NA12C-containing DMF solution after adding H_2_O. As shown in Figure 5d, the aggregates formed immediately after adding water were similar to those in Figure 2. Moreover, after 2 h of adding H_2_O, the aggregates became rod-shaped (chip) with bright ends, more uniform and brighter. The SEM images show the different shapes and sizes of the aggregates. (Figure 5e,f) Similar results were also observed in other ratio solutions (*f*_W_ = 80, 90%), but it took more time (>2 h).

## 3. Discussion

**Micelle-like nanoaggregate model**: The Lippert-plot results in Appendix A show that the solvent polarity has a great influence on the emission intensity and wavelength of the ICT molecule. Interestingly, in this manuscript, the AIEE properties of the nanoaggregate also present significant changes with the varying polarity of solvents. In DMF/H_2_O solution, the aggregation state gives rise to an obvious increase in quantum yields compared with that in the THF/H_2_O, with a remarkable AIEE effect. The main factor in this phenomenon is the polarity-dependent characteristics of the AIEE property of D-A fluorophore. It is known that, as an apolar molecule, its AIEE product is essentially not influenced by solvent polarity. Alternatively, reports of the literature supported a mechanism for AIEE induced by the restriction of the transition from the local excited (LE) state to the ICT state, which was proposed to account for the AIEE product with an apparent solvent effect [39,40]. In this manuscript, we propose a model that can be described in Figure 6. In zone 1, the emission of NAXC mainly comes from the LE state. As the polarity increases, the fraction of the ICT state increases obviously with the reduction in the LE state fraction. In a 40% DMF/H_2_O mixture solution, the ICT state becomes dominant, and the emission peaks of dyes such as NAXC with a strong ICT capability are greatly red-shifted, which exhibits an apparent solvent polarity effect and results in an extremely low emission intensity.

It is clear that when the *f*_W_ is less than 40%, all the NAXC compounds have similar spectral behaviors (Lippert-plot slope). This indicates that, under this condition (zone 1), the spectral performance is limited to the core scaffold and has nothing to do with the later chain, and there is no aggregation of molecules under this condition. However, in the zone 2 region (*f*_W_ 40~50%), there is a molecular alignment transition for achieving the micelle-like nanoaggregate. It can be seen from Figure 4 that the weak fluorescent emission continues to occur, and, according to the discussion of zone 3, the core must be in the outer layer to present a solvent effect, as in zone 1. Thus, we propose the micelle-like nanoaggregate model with an “alkoxyl chain inside, naphthalic anhydride outside”. That is, the morphological altering from amorphous to aggregation (head-to-head alignment) may be responsible for the unusual blue shifts at zone 2. This inference can be supported by the morphology of the nanoaggregates in the different solvent mixtures.

As described above, the molecules will become progressively less soluble, with *f*_W_ in the mixing solution increased above 50% (zone 3). Subsequently, these molecules begin to aggregate to form a cluster or micelle. We proposed that the “alkoxyl chain inside, naphthalic anhydride outside“ micelle is the most likely molecular packing structure for keeping the intrinsic solvent polarity effect. That is why the emission peaks of NA4C and NA6C are red-shifted again, as in zone 1, from those in the more aqueous mixture. The complete nano-aggregates need more H_2_O to drive and then become more compact, which can restrict the TICT of molecules on the nanoaggregate. There are three reasons to support our micelle-like nanoaggregate theory. First, we can collect the CMC for the NA2C~NA12C, as Appendix A shows. Second, the spectra switch for NA12C is quick and stable and does not need to exceed H_2_O to drive. Third, the final emission intensities of NA4C and NA6C (at *f*_W_ = 90) are close to those of NV12C, indicating that they should be the same AIEE mechanism. Hence, we inferred that the AIEE of NAXC comes from the micelle-like packing of a molecule over the suitable *f*_W_.

Compared to that of the monomer emission (~515 nm in Figure 5c), the blue-shift emission band of 505 nm is sharp and may originate from micelle-like aggregates. After 30 min, the emission intensity remains unchanged, but the emission wavelength is further blue-shifted to ~470 nm, which can be attributed to larger aggregates with a further molecule packing. According to the spectral data and discussion above, we consider that NV12C might undergo a self-assembly process in DMF/H_2_O mixed solvents, with the *f*_W_ being more than 50%, to produce the micelle-like (small) fluorescent aggregates (green). This phenomenon appeared immediately after adding water, so it should be the **kinetic AIEE** of NA12C. Then, the micellar aggregates grow further with time, forming lamellar and then leading to the formation of nanochips, which should belong to the **thermal dynamic AIEE** stacking of NA12C.

## 4. Materials and Methods

### 4.1. Materials

All the reactants in the synthesis experiment are of reagent grade, purchased from various brands and used directly. The column chromatography for purification uses Silica gel 60F 230–240 mesh ATSM Merk. Thin-layer chromatography (TLC) is used for analysis, using Silica gel 60 F245 Merck sheet TLC; after the sheet is unfolded, it is inspected with ultraviolet light. The solvent used for spectroscopy is HPLC grade and purified by standard procedures.

### 4.2. Apparatus

A Varian Mercury 400 nuclear magnetic resonance spectrometer was used as the Nuclear Magnet Resonance Spectrometer (NMR). The used solvents were deuterated chloroform (CDCl_3_) and deuterated dimethyl sulfoxide (DMSO-d6). The spectral chemical shift (δ) of hydrogen nuclear magnetic resonance spectroscopy (^1^H-NMR) is in ppm, and the chemical shift of the solvent is used as the standard. A Thermo Genesys 6 UV-Visible Spectrophotometer was used for the absorption spectra. The fluorescence spectra were measured via Jobin Yvon fluorolog-3. The fluorescence images were taken via a Leica AF6000 fluorescent microscope and Leica DFC310 FX Digital color camera (CCD). The JEOL JSM-7800F Prime Schottky Field Emission Scanning Electron Microscope from the College of Engineering NCHU was used for SEM images.

### 4.3. Synthesis

Figure 1 illustrates the synthetic routes of the 1,8-naphthalic anhydride derivatives NAXC, X = 1, 2, 4, 6, 12. For the synthesis process and identification of the intermediates 2a, 4a, 6a, 12a and 1b, 2b, 4b, 6b, 12b, please refer to the Appendix A.

The general procedure for the synthesis of NAXC final products: The high-pressure bottle was purged with nitrogen for about 1 min; then, Pd(OAc)_2_ (8 mg) and (o-tol)_3_P (80 mg) were added and then mixed with 4-bromo-1,8-naphthalic anhydride (0.42 g, 1.5 mmol) and CH_3_COOK (0.58 g, 6 mmol) in the 12 mL of the solvent pair H_2_O/1,4-dioxane = 1/5 (*v*/*v*), stirring evenly with a magnet. Finally, 2.25 mmol of 1b, 2b, 4b, 6b, 12b was added, respectively, and then bubbled with nitrogen for about 2 min. After refluxing for 2 days, the mixture was cooled down to room temperature. It was extracted twice with CH_2_Cl_2_/H_2_O, and the organic layer was dried over anhydrous magnesium sulfate (MgSO_4_) and purified via silica column chromatography (the eluent was n-hexane) to obtain a yellow solid. Appendix A show the ^1^H NMR spectrum (up) and ^13^C NMR spectrum (down) in CDCl3. 

4-(1,8-naphthalic anhydride-4-yl) anisole (**NA1C**): The yield is about 68%. ^1^HNMR (400 Hz, CDCl_3_): δ = 8.64 (d, *J* = 7.6 Hz, 2H), 8.42 (d, *J* = 8.5 Hz, 1H), 7.81–7.70 (M, 2H), 7.45 (d, *J* = 8.7 Hz, 2H), 7.10 (d, *J* = 8.7 Hz, 2H), 3.92 (s, 3H) ppm. ^13^C NMR (400 MHz, CDCl_3_) δ 161.02, 160.77, 160.45, 148.60, 134.49, 133.47, 133.24, 131.25, 130.54, 130.46, 128.38, 127.33, 119.08, 117.31, 114.53, 55.65. Melting point (mp): 145–147 °C. Exact Mass: 304.07; HRMS (ESI, *m*/*z*): [M]^+^ 303.98. EA: Anal. Calcd. For C_19_H_12_O_4_: C, 74.99; H, 3.97 (%). Found: C, 74.86; H, 3.95 (%).

1-ethoxy-4-(1,8-naphthalic anhydride-4-yl)benzene (**NA2C**): The yield is about 70%. ^1^HNMR (400 Hz, CDCl_3_): δ = 8.65 (d, *J* = 7.6 Hz, 2H), 8.43 (d, *J* = 8.5 Hz, 1H), 7.79–7.71 (m, 2H), 7.44 (d, *J* = 8.7 Hz, 2H), 7.08 (d, *J* = 8.7 Hz, 2H), 4.15 (q, *J* = 7.0 Hz, 2H), 1.49 (t, *J* = 7.0 Hz, 3H) ppm. ^13^C NMR (400 MHz, CDCl_3_) δ 161.08, 160.83, 159.86, 148.69, 134.54, 133.50, 133.28, 131.26, 130.58, 130.32, 128.39, 127.33, 119.14, 117.33, 115.04, 63.91, 15.01. Melting point (mp): 151–154 °C. Exact Mass: 318.09; HRMS (ESI, *m*/*z*): [M]^+^ 318.33. EA: Anal. Calcd. For C_20_H_14_O_4_: C, 75.46; H, 4.43 (%). Found: C, 75.32; H, 4.45 (%).

1-butoxy-4-(1,8-naphthalic anhydride-4-yl)benzene (**NA4C**): The yield is about 66%. ^1^HNMR (400 Hz, CDCl_3_): δ = 8.64 (d, *J* = 7.6 Hz, 2H), 8.43 (d, *J* = 8.5 Hz, 1H), 7.81–7.70 (m, 2H), 7.44 (d, *J* = 8.7 Hz, 2H), 7.09 (d, *J* = 8.7 Hz, 2H), 4.07 (t, *J* = 6.5 Hz, 2H), 1.84 (dt, *J* = 14.4, 6.5 Hz, 2H), 1.57–1.50 (m, 2H), 1.02 (t, *J* = 7.4 Hz, 3H). ^13^C NMR (400 MHz, CDCl_3_) δ 161.04, 160.79, 160.06, 148.72, 134.54, 133.46, 133.25, 131.23, 130.54, 130.22, 128.36, 127.30, 119.07, 117.24, 115.04, 68.11, 31.43, 19.43, 14.03. Melting point (mp): 148–151 °C. Exact Mass: 346.12; HRMS (ESI, *m*/*z*): [M]^+^ 346.53. EA: Anal. Calcd. For C_22_H_18_O_4_: C, 76.29; H, 5.24 (%). Found: C, 75.89; H, 5.27(%).

1-hexyloxy-4-(1,8-naphthalic anhydride-4-yl)benzene (**NA6C**): The yield is about 60%. ^1^HNMR (400 Hz, CDCl_3_): δ = 8.63 (d, *J* = 7.6 Hz, 2H), 8.43 (d, *J* = 8.5 Hz, 1H), 7.80–7.68 (m, 2H), 7.44 (d, *J* = 8.7 Hz, 2H), 7.09 (d, *J* = 8.7 Hz, 2H), 4.06 (t, *J* = 6.5 Hz, 2H), 1.85 (p, *J* = 6.7 Hz, 2H), 1.58–1.45 (m, 2H), 1.45–1.32 (m, 4H), 0.93 (t, *J* = 6.7 Hz, 3H). ^13^C NMR (400 MHz, CDCl_3_) δ 161.01, 160.75, 160.05, 148.71, 134.54, 133.44, 133.23, 131.23, 130.52, 130.19, 128.35, 127.29, 119.03, 117.20, 115.03, 68.42, 31.74, 29.36, 25.90, 22.78, 14.21. Melting point (mp): 139–141 °C. Exact Mass: 374.15; HRMS (ESI, *m*/*z*): [M]^+^ 373.89. EA: Anal. Calcd. For C_24_H_22_O_4_: C, 76.99; H, 5.92 (%). Found: C, 76.25; H, 5.97 (%).

1-dodecyloxy-4-(1,8-naphthalic anhydride-4-yl)benzene (**NA12C**): The yield is about 50%. ^1^HNMR (400 Hz, CDCl_3_): δ = 8.64 (d, *J* = 7.6 Hz, 2H), 8.43 (d, *J* = 8.5 Hz, 1H), 7.81–7.70 (m, 2H), 7.44 (d, *J* = 8.7 Hz, 2H), 7.08 (d, *J* = 8.7 Hz, 2H), 4.06 (t, *J* = 6.5 Hz, 2H), 1.85 (p, *J* = 6.6 Hz, 2H), 1.56–1.44 (m, 2H), 1.28 (s, 16H), 0.88 (t, *J* = 6.8 Hz, 3H). ^13^C NMR (400 MHz, CDCl_3_) δ 161.10, 160.77, 160.07, 148.72, 134.54, 133.45, 133.24, 131.23, 130.54, 130.21, 128.36, 127.30, 119.08, 117.24, 115.04, 68.44, 32.08, 29.77, 29.57, 29.52, 29.40, 26.23, 22.85, 14.29. Melting point (mp): 94–97 °C. Exact Mass: 458.25; HRMS (ESI, *m*/*z*): [M]^+^ 457. 67. EA: Anal. Calcd. For C_30_H_34_O_4_: C, 78.57; H, 7.47 (%). C_30_H_34_O_4_·1H_2_O: C, 75.60; H, 7.61. Found: C, 75.76; H, 7.63 (%).

## 5. Conclusions

The donor-acceptor (D-A) type fluorophores, NAXC with alkoxyl substituents of varying carbon chain lengths (X = 1, 2, 4, 6, 12), revealed unusual fluorescence enhancement in water. We designed NAXC as the structure–activity relationship (SAR) of the surfactant to demonstrate that AIEE came from the formation of micelle-like nanoaggregates, which caused the restriction of the transfer from the LE state to the ICT state, resulted in a blue-shift in emission and enhanced the intensity in the aggregate state. That is, the micelle-like nanoaggregate platform showed different degrees of self-assembly behaviors and corresponding aggregation-induced emission enhancement (AIEE) progresses. At the same time, different nanostructures and corresponding spectral changes can be obtained by adjusting the water ratio in the mixed solution. That is, NAXC compounds present different transitions between LE, ICT and AIEE based on the polarity and water ratio. Among them, NA12C is most likely to form a micelle and has the most obvious fluorescence enhancement, which will switch over time due to the nano-aggregation transition. The combination of ICT and AIEE properties has found utility in sensing applications, where the materials exhibit enhanced emission in response to specific analytes or stimuli. In our study, the synthesized material exhibits enhanced emission in response to the changes in the solvent polarity and time. At first, we can prepare a variable color of fluorescent power from a low-emission-yield organic molecule. Alternatively, we can obtain the bright nanostructure by the Stöber method, silica-coating the as-prepared nanoaggregate in this manuscript. These bright nanostructures can undergo further surface modification by conjugating another fluorescence sensor to fabricate the binary fluorescent nanostructure for a ratiometric sensor.

## Data Availability

Not applicable.

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
