# Peer review of "Tunable Fluorescence via Self-Assembled Switching of AIE-Active Micelle-like Nanoaggregates"

_ijms, 2023, doi:10.3390/ijms24129941_

Round 1

Reviewer 1 Report

In this paper, the authors have investigated the synthesis and characterization of the organic nano aggregates for emission switching. It is an interesting topic for the synthesis and properties of the fluorescence behaviors using self-assembly of the organic naphthalic anhydride chromophore substituted 4-alkoxyphenyl moiety. However, the important information to make the achievement of these fields is insufficient. Therefore, the authors need the revision the manuscript for publication in International Journal of Molecular Science. The question and suggestions are as followed;

We believe that the title of this manuscript is too general. We suggest that the authors can change the title of this manuscript to show the specific research topic.

The Introduction section is too weak. We suggest that authors should add more background information and related literature in the Introduction section.

The authors showed the assignment of the proton nuclear magnetic resonance (1H NMR) in the supplementary information. We suggest that the authors should add the 1H NMR spectrum of the synthesized materials including peak area to increase the understanding of this manuscript by journal readers.

This manuscript has too many typographic errors. We suggest that authors should correct typographic errors such as case sensitive, superscript, subscript, Italic issues, and spacing words in the whole manuscript.

We judged that authors should cite the related literature including recent studies in the References section. Also, the authors should add more discussion in the Results and Discussion section.

This manuscript has too many typographic errors. We suggest that authors should correct typographic errors such as case sensitive, superscript, subscript, Italic issues, and spacing words in the whole manuscript.

Author Response

Response to Reviewer #1:

Comments and Suggestions for Authors

In this paper, the authors have investigated the synthesis and characterization of the organic nano aggregates for emission switching. It is an interesting topic for the synthesis and properties of the fluorescence behaviors using self-assembly of the organic naphthalic anhydride chromophore substituted 4-alkoxyphenyl moiety. However, the important information to make the achievement of these fields is insufficient. Therefore, the authors need the revision the manuscript for publication in International Journal of Molecular Science. The question and suggestions are as followed;

  • We believe that the title of this manuscript is too general. We suggest that the authors can change the title of this manuscript to show the specific research topic.

Reply: Thanks for the reviewer’s comments and we appreciate your suggestion to edit the title. We have changed the title as follows: “Tunable fluorescence via self-assembled switching of AIE-active micelle-like nanoaggregates

  • The Introduction section is too weak. We suggest that authors should add more background information and related literature in the Introduction section.

Reply: Thanks for the reviewer’s comments. We have added a section about regulation between ICT and AIEE, and related literature in the Introduction section (second section).

  • The authors showed the assignment of the proton nuclear magnetic resonance (1H NMR) in the supplementary information. We suggest that the authors should add the 1H NMR spectrum of the synthesized materials including peak area to increase the understanding of this manuscript by journal readers.

Reply: Thanks for the reviewer’s comments. The 1H NMR spectra of the synthesized compounds have been added please refer to Figure S9-S13.

  • This manuscript has too many typographic errors. We suggest that authors should correct typographic errors such as case sensitive, superscript, subscript, Italic issues, and spacing words in the whole manuscript.

Reply: Thanks for the reviewer’s comments. We carefully revised the whole manuscript and edited all of the typographic errors, as shown in blue color

  • We judged that authors should cite the related literature including recent studies in the References section. Also, the authors should add more discussion in the Results and Discussion section.

Reply: Thanks for the reviewer’s comments. We added some of the most recent 2023 literatures to the References section (ref 21-23). Also, we added more information to the Discussion section to discussed the mechanism in detail.

Comments on the Quality of English Language

This manuscript has too many typographic errors. We suggest that authors should correct typographic errors such as case sensitive, superscript, subscript, Italic issues, and spacing words in the whole manuscript.

Reply: Thanks for the reviewer’s comments. We carefully revised the whole manuscript and edited all of the typographic errors. -----marked with blue color in the manuscript.

Reviewer 2 Report

I carefully read the manuscript "Tunable Aggregation-Induced Emission Switching of Organic Nanoaggregates" by Elsyed et al., in which they report new "NAxC" dyes as potential fluorophores.

Several spelling errors throughout the manuscript must be corrected, and, for example, "H2O" must be with the 2 in the subscript.

The beginning of the discussion could be more understandable, showing some carelessness.

My most considerable criticism focuses on the characterization of the compounds. The authors do not state whether these compounds are new. Furthermore, essential elements for their description are not presented, such as their melting points, infrared spectroscopy signals, and carbon NMR. Moreover, these spectra must be inserted in the supplementary material, not just the signals in the manuscript. These are essential data for the author to know about the purity of the dyes and the expected signs in the event that they want to reproduce some work.

Also, in characterization, there are several oversights concerning the aesthetic quality of the manuscript, which must be improved.

In the conclusions, the authors are asked to add possible future directions for the work they will develop based on the results described here.

Author Response

Response to Reviewer #2:

Comments and Suggestions for Authors

I carefully read the manuscript "Tunable Aggregation-Induced Emission Switching of Organic Nanoaggregates" by Elsyed et al., in which they report new "NAxC" dyes as potential fluorophores.

  • Several spelling errors throughout the manuscript must be corrected, and, for example, "H2O" must be with the 2 in the subscript.

Reply: Thanks for the reviewer’s comments. We carefully revised the whole manuscript and rewritten "H2O" with the 2 in the subscript (H2O).

  • The beginning of the discussion could be more understandable, showing some carelessness.

Reply: same as query 5 of reviewer #1

  • My most considerable criticism focuses on the characterization of the compounds. The authors do not state whether these compounds are new. Furthermore, essential elements for their description are not presented, such as their melting points, infrared spectroscopy signals, and carbon NMR. Moreover, these spectra must be inserted in the supplementary material, not just the signals in the manuscript. These are essential data for the author to know about the purity of the dyes and the expected signs in the event that they want to reproduce some work.

Reply: Thanks for the reviewer’s comments. We have supported the melting point data and 13C NMR spectra data to the section 4.3 in 4. Materials and Methods. Meanwhile, the 13C NMR spectra of the synthesized compounds have been added please refer to Figure S9-S13. We use the data of Raman spectroscopy instead of the data of IR spectroscopy, and it is shown in Figure S14 in supporting information.

  • Also, in characterization, there are several oversights concerning the aesthetic quality of the manuscript, which must be improved.

Reply: Thanks for the reviewer’s comments. We have modified the manuscript.

  • In the conclusions, the authors are asked to add possible future directions for the work they will develop based on the results described here.

Reply: Thanks for the reviewer’s comments. the synthesized materials could be improved and to be utilized in some applications. We have added the possible applications of as-prepared compound in this manuscript, as shown in CONCLUSION.

Reviewer 3 Report

This work reports on unique fluorescence behavior of 1,8-naphthalic anhydride derivatives in aqueous organic solvents. The authors synthesized the naphthalic anhydride derivatives NAxC with alkoxy substituents of varying carbon chain length by the Suzuki coupling reaction using palladium catalysts. The authors evaluated the fluorescence properties of the NaxC in various media including aqueous organic solvents and revealed that the NaxC, x=4, 6,12 showed unusual fluorescence enhancement in water-rich DMF. The authors also presented a model to illustrate the transitions between LE – ICT – AIEE based on the polarity, water ratio, and time changes.

This manuscript may be useful for scientists working not only with fluorophores but also with palladium catalysts for the Suzuki coupling reaction.

Some minor revision is necessary before the manuscript can be accepted.

1.     P.5, Table 1, EA

The Δf value of EA may be an outlier. Please check it.

2.     P.6 Figure 2, NA12C

‘NA12C (e)’ may be fixed to ‘NA12C (f)’.

Author Response

Response to Reviewer #3:

Comments and Suggestions for Authors

This work reports on unique fluorescence behavior of 1,8-naphthalic anhydride derivatives in aqueous organic solvents. The authors synthesized the naphthalic anhydride derivatives NAxC with alkoxy substituents of varying carbon chain length by the Suzuki coupling reaction using palladium catalysts. The authors evaluated the fluorescence properties of the NaxC in various media including aqueous organic solvents and revealed that the NaxC, x=4, 6,12 showed unusual fluorescence enhancement in water-rich DMF. The authors also presented a model to illustrate the transitions between LE – ICT – AIEE based on the polarity, water ratio, and time changes.

This manuscript may be useful for scientists working not only with fluorophores but also with palladium catalysts for the Suzuki coupling reaction.

Some minor revision is necessary before the manuscript can be accepted.

  1. P.5, Table 1, EA

The Δf value of EA may be an outlier. Please check it.

Reply: Thanks for the reviewer’s comments. We have checked the value and we found that it was wrong. We added the correct value to the table.

  1. P.6 Figure 2, NA12C

‘NA12C (e)’ may be fixed to ‘NA12C (f)’.

Reply: Thanks for the reviewer’s comments. We fixed the figure caption to ‘NA12C (f)’

Round 2

Reviewer 1 Report

This manuscript in current form is acceptable.

This manuscript in current form is acceptable.

Reviewer 2 Report

The manuscript has been very significantly improved. The NMR data show the purity of the compounds, and all other data complement the work, making it much more appealing and enriched.

Considering the fact that most of my comments were addressed, I consider the manuscript ready for publication.